# A Novel Miniature and Selective CMOS Gas Sensor for Gas Mixture Analysis—Part 3: Extending the Chemical Modeling

**DOI:** 10.3390/mi14020270

**Published:** 2023-01-20

**Authors:** Boris Goikhman, Moshe Avraham, Sharon Bar-Lev, Sara Stolyarova, Tanya Blank, Yael Nemirovsky

**Affiliations:** Electrical and Computer Engineering Department, Technion—Israel Institute of Technology, Haifa 32000, Israel

**Keywords:** gas sensor, pellistor, CFD, TMOS, GMOS

## Abstract

This is the third part of the paper presenting a miniature, combustion-type gas sensor (dubbed GMOS) based on a novel thermal sensor (dubbed TMOS). The TMOS is a micromachined CMOS-SOI transistor, which acts as the sensing element and is integrated with a catalytic reaction plate, where ignition of the gas takes place. The first part was focused on the chemical and technological aspects of the sensor. In Part 2, the emphasis was on the physical aspects of the reaction micro-hot plate on which the catalytic layer is deposited. The present study focuses on applying several advanced simulation tools, which extend our understanding of the GMOS performance, as well as pellistor sensors in general. The three main challenges in simulating the performance are: (i) how to define the operating temperature based on the input parameters; (ii) how to measure the dynamics of the temperature increase during cyclic operation at a given duty cycle; (iii) how to model the correlation between the operating temperature and the sensing response. The simulated and analytical models and measured results are shown to be in good agreement.

## 1. Introduction

Ongoing technological progress provides a strong demand for an accurate description of air composition. Agriculture [1], safety [2], security [3] and IoT [4] require an accurate and rapid detection of the wide spectrum of volatile organic compounds (VOCs). The novel pellistor-like gas sensor dubbed GMOS is a low-power portable gas sensor with a rapid response time of a few seconds and a sensitivity of the sub-ppm range. In previous works, the sensing [5] and electrical [6] properties of this sensor were described. This work focuses on the investigation of the sensing mechanism, modeling the thermodynamic and chemical processes lying in its basis, by applying advanced tools for a 3D simulation of fluid dynamics.

A GMOS gas sensor operates by the pellistor principle. The sensing element—a highly sensitive temperature sensor dubbed TMOS—detects the heat of combustion of the volatile organic compounds (VOCs) over the reacting surface covered with Pt catalyst. The sensing surface is heated with a meander heater, which defines the range of operating temperatures up to 300 °C based on the input voltage.

The operating temperature is an important feature of many gas sensors. While there are works reporting the room-temperature sensors [7], most of the gas-analytical systems rely on the operating temperatures in the 100–400 °C range [8,9]. The precise detection and control of the operating temperature is crucial for the understanding of the sensor, as this is one of the main parameters defining the sensing response [10]. The task of measuring the operating temperature can be challenging, as it requires additional equipment, such as IR sensors. The operating principle of the GMOS sensor allows the avoidance of these complications. The highly sensitive temperature sensor (TMOS) implemented in the GMOS setup as a sensing element can be also used as the thermometer for operating temperature detection. 

In this article, the computational fluid dynamics (CFD) model of the GMOS sensing element is reported. The first part of the article is devoted to the establishment of the correlation between the heater input voltage and the operating temperature of the sensing surface. The sensor heating was modeled with ANSYS Fluent software [11]. The predicted data of sensor heating and sensing performance have shown a remarkable correlation with the experimental results. 

The main part of this work addresses the investigation of the sensing mechanism of the GMOS sensor. Evaporated ethanol was used as the exemplary target gas. The correlation between the sensing response and the operating temperature was modeled, simulated, and compared with the experimental data. 

The important property of the operating sensor is its response time. For the GMOS sensor, the response time is defined by the sampling rate of the electrical circuit and by the time of system thermal stabilization. In this study, the thermal stabilization of the sensor and air around it was simulated with ANSYS Fluent, considering the natural convection, sensor thermal capacity and combustion of ethanol vapors [11]. The thermal time constant was established with the 3D simulation of the sensor thermal stabilization during the first 40 ms of heating. 

## 2. Materials and Methods

### 2.1. The GMOS Sensor

GMOS is a portable low-power pellistor-type sensor (Figure 1a). TMOS—the temperature sensing element of GMOS—is a highly sensitive micromachined transistor-based temperature sensor [12,13,14]. The TMOS sensor is a MOSFET transistor operating at subthreshold, highly sensitive to temperature conditions. The GMOS sensing principle is based on the detection of the heat of the reaction released during the VOC combustion above the sensing surface covered with Pt catalyst. The temperature of the sensing surface is regulated, with the meander heater deposited below it (Figure 1b). While various approaches were applied to find the correlation between the heater input voltage and the heater temperature [6], in this study the CFD modeling of the sensor heating is applied, considering the thermal exchange with the surrounding air [11]. The heater input voltage lies in the 0.5 V–3.0 V range, and the heater resistance is designed for 600 Ohm at room temperature. 

The sensing layer of the GMOS sensor Is covered with the Pt catalyst (Fraunhofer PT-LT-20 20 wt. % nano-particle ink for aerosol jet printing, IKTS Dresden Lot no Pt073/January 2019). The 1:2 solution of the catalyst in thinner (Solvent Mixture, co-solvent for Pt-Lt-20 inks IKTS Dresden/January 2019) was deposited on the sensing surface manually, then the sensor was dried at the heating plate at 200 °C for 20 min. The bonded sensor was heated at the 3.5 V heater input voltage (>300 °C) for 30 min. 

For the investigation of the sensing performance of the GMOS, the gas sensor was located inside the hermetic gas chamber (volume 6 L). The sensing signal in the atmospheric air was recorded for 25 s for four different operating temperatures. The analyte vapors were introduced by injection of the fixed amount (10 μL, corresponds to 700 ppm) of the liquid ethanol, which was evaporated inside the chamber with the ventilator. The sensing response was recorded for 100 s for each operating temperature. The sensing response was calculated as the difference of the signals averaged over the last 40 s and the first 20 s of the experiment. The experiment was conducted 3 times, providing the total sensing response for 12 operating temperatures in the range of 23 °C–270 °C.

### 2.2. Simulation

The CFD simulation of the air dynamics around the sensor was conducted with ANSYS Fluent software [11]. The sensor was modeled as a 213 × 213 × 5 ${\mu m}^{3}$ cuboid, filled with $SiO_{2}$ (density $2200\;\mathrm{kg}/m^{3}$; thermal conductivity 1.4 W/(m⋅K); normalized heat capacity 730 J/(K⋅kg)). To reduce the required computational power, the meander heater was modeled as a 200 × 200 × 1 ${\mu m}^{3}$ cuboid integrated inside the sensor and filled with tungsten (density 19,300 $\mathrm{kg}/m^{3}$; thermal conductivity 173 W/(m⋅K); normalized heat capacity 134 J/(K⋅kg)). To simulate the heating process, the energy Source Term was set as the Heater Cell Zone Condition. The fluid domain around the sensor was sized 600 × 600 × 800 ${\mu m}^{3}$ (with greater vertical dimension). The side walls were treated as a pressure-inlet for dry air (300 K) with zero pressure difference, while the top wall was considered as a pressure-outlet with zero pressure difference, which allowed the simulation of the thermal interaction with the gas chamber, saving the computational power. The ethanol flow through inlet/outlet walls was set to be zero, thus it was consumed only in the process of combustion. The mesh sizing was set in a range of 1–20 μm.

To simulate the impact of the Pt catalyst, the default pre-exponential factor and activation energy parameters of ethanol combustion were modified according to [15] (Pre-exponential factor $A=4.57\times 10^{9},$ activation energy $Ea=5.538\times 10^{7}\;J/\mathrm{kmol}$). Ethanol combustion was simulated as a volumetric reaction (“Direct Source” Chemistry solver). “Finite Rate/No TCI” turbulence-chemistry interaction algorithm was applied, as the process of the ethanol combustion was mainly reaction-driven (see Section 3.2.2). 

## 3. Results

### 3.1. Voltage–Temperature Correlation

The temperature sensing element of the GMOS sensor can be applied as a thermometer for the direct measurements of the operating temperature. The voltage–temperature correlation can be extracted from the I-V curves of the investigated GMOS sensor. Below is presented the theoretical derivation of the expression for the operating temperature, based on the I-V curves of the TMOS transistor.

As the sensing element of GMOS is the MOSFET sensor operating in subthreshold, its drain-source current $I_{ds}$ is defined by the following formula [16]:(1)$$I_{DS}=\mu \left(T\right)C_{ox}\frac{W}{L}{\left(\frac{kT}{q}\right)}^{2}\left(n-1\right)exp\left(\frac{q}{nkT}\left(V_{GS}-V_{T}\right)\right)\left(1-exp\left(-\frac{qV_{DS}}{kT}\right)\right)$$
where W and L—channel width and length; for the transistor used in the experiment $\frac{W}{L}=1425;$ $\frac{kT}{q}=0.02586\;V$ for T=300 K*;*
μ(T)—carriers mobility; $C_{ox}—$oxide thermal capacity; n—capacitance ratio; $V_{T}\;—$ threshold voltage; $V_{DS}$ and $V_{GS}—$drain-source and gate-source voltages, respectively.

The capacitance ratio is defined as $n=1+\frac{C_{s}+C_{ss}}{C_{ox}}$, where $C_{s}$—semiconductor capacitance and $C_{SS}$—fast surface states capacitance. 

For the subthreshold regime ($V_{GS}<V_{T}$) it is true that $\mu \left(T\right)=\mu_{0}{\left(\frac{T}{T_{0}}\right)}^{-2}$, thus:(2)$$I_{DS}=\mu_{0}\;C_{ox}\frac{W}{L}{\left(\frac{kT_{0}}{q}\right)}^{2}\left(n-1\right)exp\left(\frac{q}{nkT}\left(V_{GS}-V_{T}\left(T\right)\right)\right)$$

The threshold voltage $V_{T}\left(T\right)$ can be approximated with the Taylor series: (3)$$V_{T}\left(T\right)=V_{T}\left(T_{0}\right)+\frac{dV_{T}}{dT}\left(T-T_{0}\right)+\frac{1}{2}\frac{d^{2}V_{T}}{dT^{2}}{\left(T-T_{0}\right)}^{2}$$

For the n-MOS transistor fabricated in [17], $V_{T}\left(T_{0}\right)\approx 1.4\;\left[V\right]$; $\left|\frac{dV_{T}}{dT}\right|\approx -2.5\times 10^{-3}\left[V/K\right]$; and component $\frac{d^{2}V_{T}}{dT^{2}}\to 0$ for a given temperature range [18]. After the substitution:(4)$$I_{DS}=I_{0}exp\left(\frac{q}{nkT}\left(V_{GS}-V_{T}\left(T_{0}\right)+\frac{dV_{T}}{dT}\left(T-T_{0}\right)\right)\right)$$
where $I_{0}=\mu_{0}C_{ox}\frac{W}{L}{\left(\frac{kT_{0}}{q}\right)}^{2}\left(n-1\right)$.
(5)$$\log_{10}\left(\frac{I_{DS}}{I_{0}}\right)=\frac{q}{ln\left(10\right)nkT}\left(V_{GS}-V_{T}\left(T_{0}\right)+T_{0}\frac{dV_{T}}{dT}\right)+\frac{q}{ln\left(10\right)nk}\frac{dV_{T}}{dT}$$

From the I-V curves of the TMOS transistor in coordinates $log_{10}\left(I_{DS}\right)-\left(V_{GS}\right)$ (see Figure 2a), one can extract the information about the transistor temperature based on the derivative $S=\frac{d\log_{10}\left(I_{DS}\right)}{dV_{GS}}$ in the subthreshold regime:(6)$$\frac{1}{S}={\left(\frac{d\log_{10}\left(I_{DS}\right)}{dV_{GS}}\right)}^{-1}\approx \;\frac{ln\left(10\right)nkT}{q};$$

The results of I-V-measurements of the TMOS transistor are presented below. The measurements were collected for the heater voltages from the [0 V,3.0 V] range, as for higher voltages the subthreshold regime condition is no longer valid due to a high transistor temperature. 

The capacitance ratio n can be calculated from the substitution to the equation for the slope at $V_{heater}=0\;V$ at a given temperature of 300 K. For the given setup, n ≈2.11. The results of the experimental measurement of the $T_{heater}\left(V_{heater}\right)$ correlation are presented in Figure 3.

The observed results were compared to the CFD simulation. The heating of the sensing element was simulated in the atmosphere of pure dry air at 27 °C. The power of the energy source inside the heater was calculated as $P=V^{2}{}_{heater}/R_{heater}$, where $R_{heater}\approx 600\;\Omega$. The results of the simulation for the $V_{heater}\in \left[0\;V,3.2\;V\right]$ are presented in Figure 3. One can observe the great correspondence between experimental and predicted data, confirming the viability of the simulation model. 

### 3.2. Ethanol Combustion under the Different Operating Temperatures

#### 3.2.1. Theoretical

In their monumental paper, Wise et al. [15] schematically describe the heat-generation of VOCs combustion and the heat-loss flux above the hot sensing surface covered with Pt catalyst at different operating temperatures. The experimental approach of the work involves the variation of the catalyst temperature rather than the reactant concentration. The presented model (Figure 4) considers the energy fluxes resulting from heat generation by exothermic reaction (blue solid line) and heat loss by conduction-convection (red dashed lines) at the catalytic surface immersed in an atmosphere containing the VOC reactants and an inert gas. For a given concentration of reactants, the heat generation curve as a function of catalyst temperature exhibits the typical sigmoid shape. The slope of the lines representing the heat losses is defined by the atmosphere conditions, while the chemical properties of the catalytic layer and the VOC determine the shape of the sigmoid. Under the given operating temperature conditions ($T_{0x}$), the system can stabilize in multiple states, where the energy flux of heat generated during the VOC combustion equals the conduction-convection heat losses (blue solid dots). Wise et al. highlight three types of equilibrium points. The first group of stable states ($T_{rx}$) is defined by the speed of reaction of VOC combustion and is located on the left “reaction-driven” branch of the sigmoid. The third group ($T_{dx}$) is formed by the stable conditions determined by the speed of reactant diffusion toward the reaction plate. ($T_{ux}$) is located in between the group of unstable states, where the transition from the reaction-controlled to the diffusion-controlled regime occurs. The rapid transition to diffusion is defined as “thermal ignition”, often symbolized by $T^{*}$. In practice, this unstable point cannot be measured, but it may be defined by modeling. 

#### 3.2.2. Experimental

The sensing response of the GMOS sensor toward 700 ppm of ethanol mixed with atmospheric air was investigated. The sensing response was recorded in the range of operating temperatures from 23 °C to 270 °C ($V_{heater}\in \left[0\;V,3.0\;V\right]$). The heater voltages were converted to the operating temperatures according to the experimental data presented in Figure 3. The sensing response was recorded as the averaged change of the output voltage, respectively, to the baseline output voltage in the atmospheric air without ethanol. The results of the experiment are presented in Figure 5. 

The results were recorded for the heater temperatures lying below the ignition temperature of ethanol, thus forming the left, reaction-driven, branch of the sensing sigmoid of Figure 4 [15]. 

The ethanol combustion was simulated with ANSYS Fluent software [11]. The Finite-Rate Turbulence-Chemistry Interaction algorithm was applied to simulate the “reaction-driven” branch of the sigmoid. This algorithm neglects the turbulent effects of ethanol diffusion toward the reaction plate. Thus, the speed of reaction in the simulation is defined by the combustion of ethanol and not by the speed of its diffusion towards the hot plate. 

The overall trend toward exponential behavior can be seen in both experimental data and the simulation. In further works, we expect to find the experimental correlation between the heat the of reaction and the output system voltage based on the TMOS properties. The reduction of these two datasets toward the same dimension will allow the further adjustment of the input parameters of the simulation, such as the energy of activation and pre-exponential factor, allowing a deeper understanding of the thermal properties of the setup. 

#### 3.2.3. Matlab Simulation 

The analysis of a catalytic-based reaction requires the process-related catalytic parameters of the surface-controlled reaction, such as the energy of activation, the pre-exponential factor, the transport parameter, and the order of the reaction. The nature of the deposited catalytic layer depends on the specific ink, the area of the hot plate, and the sensor’s overall structure. The implementation of the Matlab simulation allows the determination of some of these parameters based on the experimental results. Moreover, the simple one-dimension model with the correct parameters of the rate constant $k_{s}$ gives us a physical insight, thus supporting and completing the simulations.

The results of the Matlab modeling of the catalytic reaction of the volumetric ethanol combustion above the sensing surface are presented in Figure 6. The simulation is based on the law of mass conservation: the surface reaction flux $F_{S}$ equals the diffusion flux $F_{D}$. The model aims to illustrate the experimental results; thus, few simplifications are applied: the corner effects are neglected (infinite one-dimensional reaction plate), as well as the temperature dependence of the diffusion coefficient, and the air is considered to be an ideal gas. In the Matlab model, the two-layer system is considered, with one layer above the catalytic plate (analyte concentration $C_{S}$), and another representing the atmospheric air with analyte concentration $C_{G}$ (Figure 6a).
$$F_{S}=\eta k_{s}\left(T\right)C_{s}$$
$$F_{D}=\frac{D}{\delta }\left(C_{G}-C_{S}\right)$$
where $\eta =Pexp\left(-E_{k}/\left(kT\right)\right)$—reaction efficiency, P—dimensionless coefficient, $E_{k}$—assumed kinetic energy of adsorbed ethanol, D—diffusion factor, and δ—width of the imaginary layer above the catalytic surface (Figure 6a). Following the approximations described above, it is easy to derive $C_{s}\left(T\right)=C_{G}{\left(1/k_{s}\left(T\right)+\eta \delta /D\right)}^{-1}$. The total power of the analyte combustion over the reacting plate can be expressed as:$$W\left(T\right)=\eta QSF_{S}\left(T\right)=\frac{\eta QSC_{G}}{1/k_{s}\left(T\right)+\eta \delta /D}$$
where Q—enthalpy of the analyte combustion, and S—area of the reacting plate. The results of the Matlab modeling of the power of reaction depending on the operating temperature are presented in Figure 6b. The parameters for the modeling were taken as follows: $D\approx 1.6\times 10^{-6}m^{2}/s$ [19], δ=150 μm, P ≈100, $E_{k}\;\approx 70\;\mathrm{kJ}/\mathrm{mole},$ and the components of the Arrhenius equation were taken similarly to ones implemented in ANSYS simulation. 

### 3.3. Time Constant Investigation

The results of the 3D simulation of the sensor heating are presented below (Figure 7). Under the initial conditions, the system lies in a thermal equilibrium of 27 °C. The initial ethanol concentration was set equal to 600 ppm. The first 40 ms of the sensor heating were simulated with a timestep of 1 ms. The power of the energy source inside the heater domain was set to be equal to the power of the heater receiving 3 V input voltage, thus the sensor temperature stabilizes at approximately 275 °C.

In Figure 7a–d, the 3D distribution of the temperature is presented over the sensing surface at the sampling of two perpendicular vertical planes, passing through the center of the sensor (thus, only a quarter of the sensor can be seen in the figure). The temperature encasement over time is presented in Figure 7n. The sensor reaches thermal stability in 20 ms. The thermal time constant, defined as the time required to reach 63.2% (e.g., 1-1/e) of the thermal equilibrium, equals 4.2 ms. 

The side view of the 3D simulation of the heat of the reaction changing over time is presented in Figure 7e–h. Current results show that the limiting factor of the thermal stabilization of the sensor is the ignition of ethanol. The reaction reaches 63.2% of its maximal power in 10.3 ms (approx. 30 ms for full stabilization). 

The side view of the simulation of ethanol distribution is shown in the third row of Figure 7i–l. The non-uniformity of the ethanol concentration is mainly defined by the thermodynamics of the process, and the equilibrium concentration difference above the sensing surface turned out to be approximately 300 ppm.

## 4. Summary

This study closes the analysis from the one-dimensional models of the first two parts of this work [5,6] with the 3D CFD analysis, provided by ANSYS Fluent simulations. The first part of this work focuses on the experimental investigation of the sensing properties of the developed GMOS sensor, as well as the sensing layer deposition techniques. The second part addresses the optimization of the electric setup design, as well as the analysis of the thermal properties of the GMOS sensor. For the theoretical justification of the sensing process, the one-dimensional physical–chemical model was implemented in the first two parts. This model, with small improvements, is presented in Section 3.2.3 of this manuscript. Being a one-dimensional model, the simulation presented in Parts 1 and 2 is quite oversimplified and does not consider the natural convection and corner effects; the linear approximation applied previously describes the surface reaction only for low concentrations of the reacting species. The temperature gradients around the hot plate are steep, and the properties of the gas vary between the reacting surface and the surrounding air. Nevertheless, this simple model still predicts the overall features observed experimentally, namely the mass-transfer and the surface-reaction controlled regions. The implementation of the advanced MATLAB simulation of the physical–chemical processes ongoing above the sensing surface allows the illustration of the general structure of the sensing process.

For the first time, the GMOS sensing process was modeled with CFD tools, which allowed the extraction of data regarding the thermal time constant, as well as the simulation of the sensing response depending on the operating temperature. The reported method may be applied to the other models of the pellistor-type gas sensors. Most of the gas sensors are used to recognize the sensing response towards the VOC, to which the sensor has already been exposed before. The prediction of the sensing response towards the unknown VOC is an important task, currently solved with ML tools, requiring collection of the large sensing databases. The described CFD model is expected to be further applied for the prediction of the sensing response towards the VOCs and its mixtures without the preliminary “training” of the sensor. 

In this work, first, the thermal heating of the sensor was investigated. The precise correlation between the heater voltage and the operating temperature was modeled and investigated experimentally. The simulated data have shown great correspondence with the experimental results, confirming the model’s validity. 

Secondly, the developed model was applied to investigate the combustion of ethanol vapors above the sensing surface. Both experimental and simulated results show the exponential growth of the sensing response in agreement with theoretical predictions. These results not only prove the correct understanding of the sensing mechanism, but can also be used for the investigation of the chemical properties of the catalytic layer. The next step toward the creation of a digital twin of GMOS will be the experimental estimation of the correlation between the sensing response and the heat of the reaction. It will allow the more precise derivation of activation energy and preexponential factor in the Arrhenius equation describing the ethanol combustion.

Finally, the thermal time constant of the sensor was investigated. The ethanol ignition was proven to be the limitational factor of the reaction, with the thermal time constant of the reaction of approximately 10.3 ms. This illustrates the rapid response time of the GMOS sensor compared to the alternatives and sheds light on some details regarding the properties of the GMOS electric circuit, as the GMOS sensor is operated by duty cycles of the applied heating voltage to reduce the power. The simulations enable us to differentiate between the thermal time constant with and without a chemical reaction. The need to apply heater voltage >40 ms to stabilize the chemical reaction was observed experimentally and is now confirmed by the CFD modeling.

## Figures and Tables

**Figure 1 micromachines-14-00270-f001:**
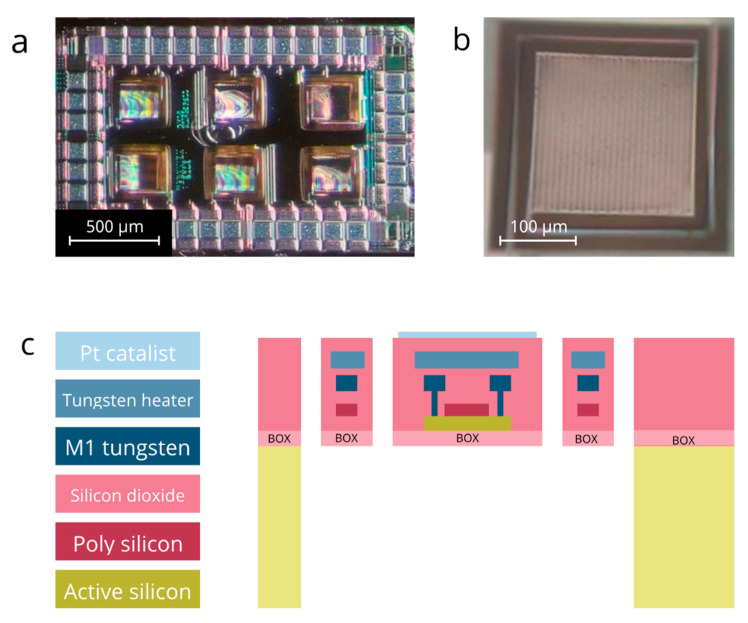
(**a**) The microscopic image of the GMOS sensing die with six sensing pixels. The die size is 2000 μm × 1380 μm; (**b**) The microscopic image of a single sensing pixel. The pixel size is 213 μm × 213 μm; (**c**) The cross-section scheme of the GMOS sensing pixel deposited above the buried oxide (BOX).

**Figure 2 micromachines-14-00270-f002:**
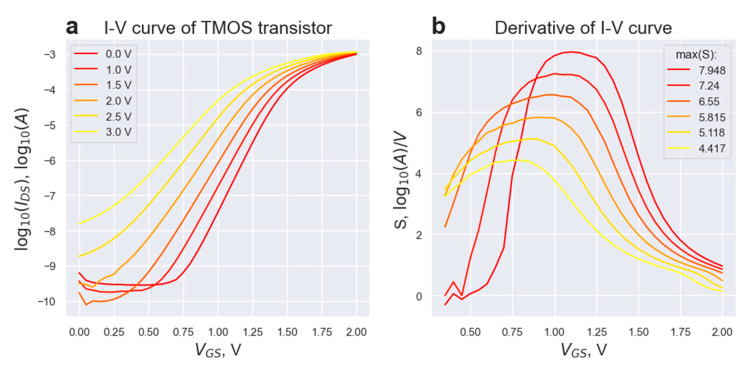
(**a**) I-V curves for the TMOS transistor at six heater voltages (0 V–3 V) at subthreshold region. (**b**) Plot of the derivative of the I-V curves $S\left(V_{GS}\right)$ = $\frac{d\log_{10}\left(I_{DS}\right)}{dV_{GS}}$ for six heater voltages; the slope at subthreshold is approximated with the slope at the point of inflection of function $\log_{10}\left(I_{DS}\right)-\left(V_{GS}\right)$.

**Figure 3 micromachines-14-00270-f003:**
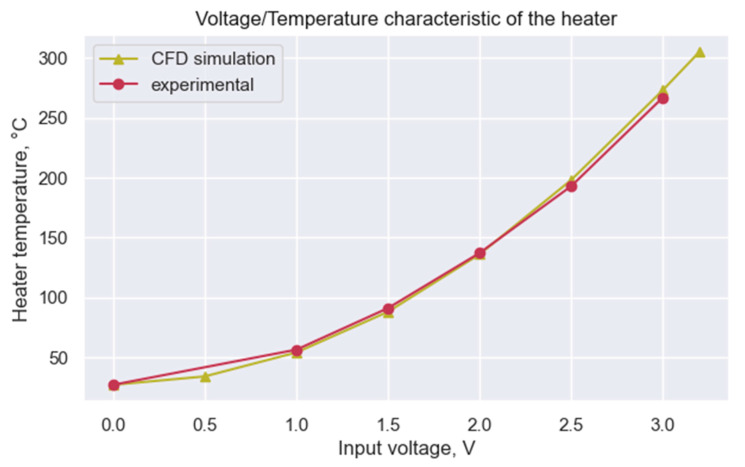
The operating temperature dependence of the heater voltage. Comparing simulated and experimental data.

**Figure 4 micromachines-14-00270-f004:**
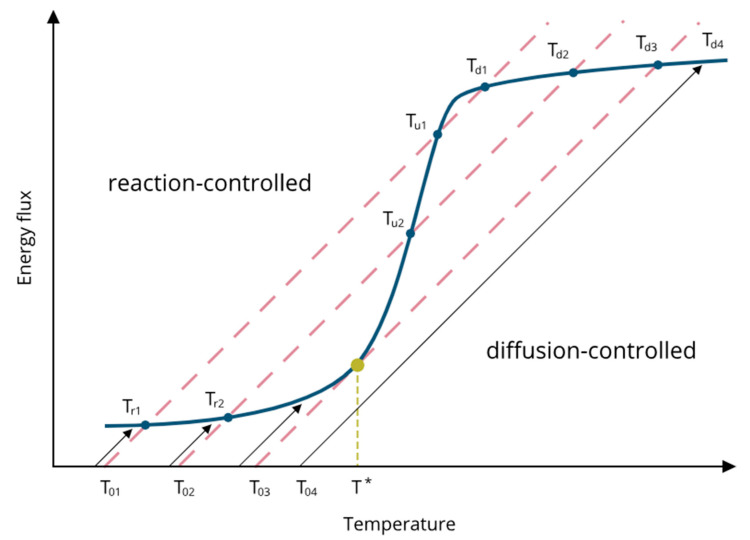
Schematic representation of heat-generation flux (blue solid line) and heat-loss flux (pink dashed line) at different operating temperatures. Ignition temperature $T^{*}$ marked with yellow [15].

**Figure 5 micromachines-14-00270-f005:**
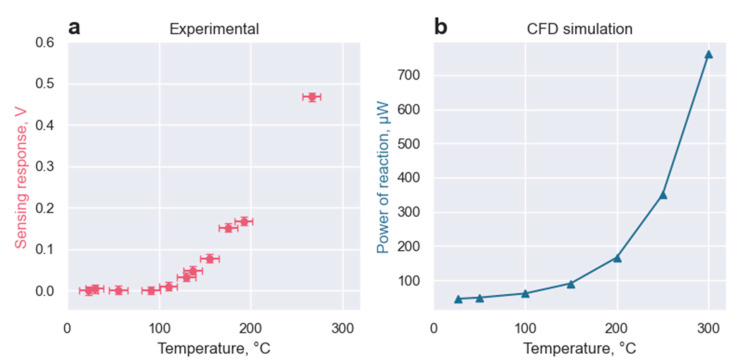
(**a**) Comparing experimental data and (**b**) simulation of the sensing response of the GMOS sensor towards 700 ppm of ethanol.

**Figure 6 micromachines-14-00270-f006:**
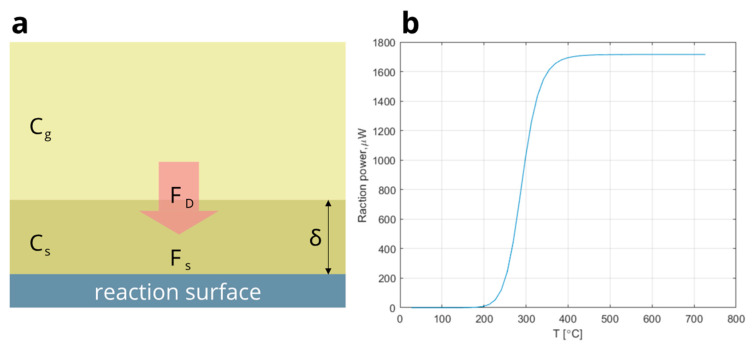
(**a**) Two-layer model of volumetric ethanol combustion; (**b**) MATLAB simulation of the sensing sigmoid.

**Figure 7 micromachines-14-00270-f007:**
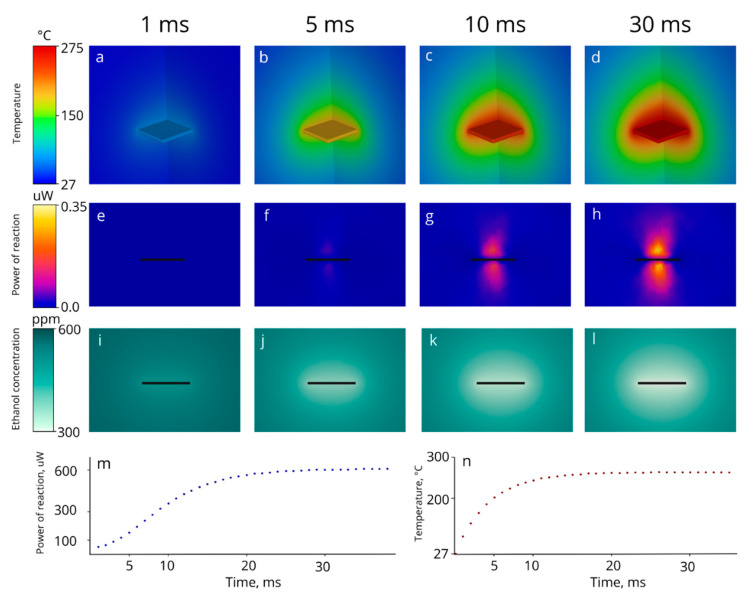
Simulation of the sensor heating in the atmosphere of dry air mixed with ethanol (600 ppm). The voltage applied to the heater is 3 V, corresponding to ~275 °C; (**a**–**d**)—the temperature distribution above the sensing surface; (**e**–**h**)—cross-section of the reaction heat distribution indicating the dynamics of the combustion reaction; (**i**–**l**): distribution of the ethanol concentration above sensing surface; (**m**)—total power of reaction of ethylene combustion changing over time; (**n**)—operating temperature changing over the time.

## Data Availability

The data presented in this study are available on request from the corresponding author.

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
