# Peer review of "A Novel Miniature and Selective CMOS Gas Sensor for Gas Mixture Analysis—Part 3: Extending the Chemical Modeling"

_micromachines, 2023, doi:10.3390/mi14020270_

Round 1

Reviewer 1 Report

The paper describes the third part of the paper presenting a miniature, combustion-type gas sensor (dubbed GMOS) based on a novel thermal sensor (dubbed TMOS). The present study focuses on applying several advanced simulation tools, which extend our understanding of the GMOS performance, as well as pellistor sensors in general.

Overall, the results are technically sound but the novelty and importance are not standing out from the discussions and summary. On the other hand, why the manuscript need to separate as many pieces with short length? In addition, the results cannot chain well with the part 1 and 2 in this manuscript. The referee recommends this manuscript not for being published in the journal micromachines.

Author Response

We appreciate the constructive comments and gladly reply to the raised questions.

To elaborate on the novelty and importance of the reported results the conclusions part was extended, please find the updated version attached, the modifications are marked with blue in the text. The separation of the work into three parts is the result of the consecutive investigation process, one can mention each part being published within two years, once the new experimental or theoretical results are acquired. At the same time, all three manuscripts address different aspects of the GMOS sensor performance (part 1 reports the sensing performance and the sensing layer deposition techniques; part 2 – the thermal properties of the sensor and the experimental protocols of the electric circuit; part 3 – compares different approaches toward the modeling of the sensing mechanism as well as derives important physical-chemical features of the sensor, such as the thermal time constant). The separation into parts here aims to address the general topic uniting these three works – the investigation of the novel GMOS sensor and its properties.

As all three manuscripts address different topics related to the GMOS sensor performance, the list of results that can chain from the previous parts is quite limited. The only intersecting topic reported is the investigation of the sensor temperature correlation with the heater input voltage. The new CFD approach is described, but the results reported in the current manuscript are in good correspondence with the ones reported in part 2.

Reviewer 2 Report

This paper is well written. It is can be accepted.

The authors only gave the simulation results of one sensor with the size of 213??x213??x5 ??. Is it the optimized design? If possible, please prove that using the simulation method.

Reviewer 3 Report

The manuscript ‘A Novel Miniature and Selective CMOS Gas Sensor for Gas Mixture Analysis—Part 3: extending the chemical modeling’ by Boris Goikhman et al. is devoted to the extended study of a previously described miniature, combustion-type gas sensor based on a thermal sensor. The authors provide improved 3D CFD analysis, the results of which are in good agreement with the experimental data. Although the sensitivity was tested only for ethanol vapors, the approach could be extended for the investigation of gas mixtures. The text of the manuscript contains some misprints (like in line 194 ‘but is may be’, etc.) which have to be corrected, but does not influence the perception of the text. The proposed manuscript could be of interest for the readers and can be recommended to be published in Micromachines.

Author Response

We are thankful for the positive feedback. The punctual and grammar misprints were corrected, and the updated version was additionally checked with Grammarly software. All the corrections are marked with yellow in the supplementary text attached.

Round 2

Reviewer 1 Report

The revised manuscript shows technically sound, and the drawn conclusion is well supported by the data. The referee recommends this manuscript for being published in the journal Micromachines as it is.